# A novel code representation for detecting Java code clones using high-level and abstract compiled code representations

**Fahmi H. Quradaa** [1,2☯]*, **Sara Shahzad** [1☯], **Rashad Saeed** [1,2☯], **Mubarak M. Sufyan** [1,3]

**1** Department of Computer Science, University of Peshawar, Peshawar, Pakistan, **2** Department of Computer Science, Aden Community College, Aden, Yemen, **3** Department of Networks and Cyber Security, AlJanad University Of Science and Technology, Taiz, Yemen

☯ These authors contributed equally to this work.
* Qurada@uop.edu.pk

**Data Availability Statement:** The dataset utilized in the present study can be accessed at the following link: https://github.com/clonebench/

## Abstract

In software development, it's common to reuse existing source code by copying and pasting, resulting in the proliferation of numerous code clones—similar or identical code fragments—that detrimentally affect software quality and maintainability. Although several techniques for code clone detection exist, many encounter challenges in effectively identifying semantic clones due to their inability to extract syntax and semantics information. Fewer techniques leverage low-level source code representations like bytecode or assembly for clone detection. This work introduces a novel code representation for identifying syntactic and semantic clones in Java source code. It integrates high-level features extracted from the Abstract Syntax Tree with low-level features derived from intermediate representations generated by static analysis tools, like the Soot framework. Leveraging this combined representation, fifteen machine-learning models are trained to effectively detect code clones. Evaluation on a large dataset demonstrates the models' efficacy in accurately identifying semantic clones. Among these classifiers, ensemble classifiers, such as the LightGBM classifier, exhibit exceptional accuracy. Linearly combining features enhances the effectiveness of the models compared to multiplication and distance combination techniques. The experimental findings indicate that the proposed method can outperform the current clone detection techniques in detecting semantic clones.

## 1 Introduction

Reusing existing source code via copy and pasting, instead of rewriting a similar code from scratch, is a common practice in software development. This practice is favored for its convenience and its capacity to significantly reduce the time and effort required for software development [1]. However, this practice often leads to the emergence of similar or identical code fragments within the software system, typically termed code clones [2]. Previous research indicates that approximately 7% to 23% of the source code within a typical software system can be identified as cloned code [3, 4].

BigCloneBench?tab=readme-ov-file and support file S4.

**Funding:** The author(s) received no specific funding for this work.

**Competing interests:** The authors have declared that no competing interests exist.

Researchers categorize code clones into four types [5, 6]. Types I, II, and III focus on textual similarities, while Type IV emphasizes functional or semantic resemblances. Type I clones are nearly identical, differing only in minor aspects like comments and formatting. Type II clones are syntactically identical with minor variations in identifiers and literals. Type III clones are syntactically equivalent with additional modifications. Type IV clones exhibit substantial dissimilarities in both text and syntax. Within spectrum of Type III and IV clones lies the "Twilight zone" [7], wherein clones are further classified based on syntactic similarity percentages as follows: Very Strongly Type-III (VST3) (90% to less than 100%), Strongly Type-III (ST3) (70% to less than 90%), Moderately Type-III (MT3) (50% to less than 70%), and Weakly Type-III/Type-IV (WT3/4) (less than 50%).

While code clones can accelerate the software development process, they have an inherent adverse impact on software quality, particularly concerning the maintainability and comprehensibility of source code [8]. Code clones can engender the propagation of bugs, irregularities in updates, inflated codebases, and contribute to the erosion of software architecture [9]. Consequently, many approaches for detecting code clones have been proposed in the literature. Generally, these existing techniques fall into six primary categories: text-based, token-based, syntax-based, semantic-based, metrics-based, and hybrid clone detection techniques [5, 10–12].

In the clone detection technique, the choice of source code representation serves as a critical factor that not only defines the upper limit for information extraction but also influences the model design which ultimately affects the final performance [13]. While many existing techniques primarily emphasize high-level representations of source code, they often neglect the corresponding low-level intermediate representations, like bytecode, assembly or other intermediary forms. Interestingly, source code fragments exhibiting syntactical differences but performing similar functions can produce comparable low-level intermediate representations. This suggests that intermediate representations (IRs) may uncover potential semantic clones, which are typically challenging to identify when relying only on high-level source code representations [14]. Therefore, the choice of code representation plays a crucial role in enhancing the efficiency of code clone detection techniques, especially in the identification of semantic clones [15]. Effective code representation should, therefore, facilitate the extraction of comprehensive syntactic and semantic features, ultimately improving the detection of both syntactic and semantic clones. Hence, there exists a pressing need for a more efficient method of representing code fragments to enable the extraction of comprehensive syntactic and semantic features.

In this work, a novel code representation is proposed to enhance the detection of semantic clones in Java source code. This approach utilizes machine learning techniques to more effectively identify similarities between clones. It integrates syntactic features extracted from the high-level representation of source code, the abstract syntax tree (AST), with semantic features derived from low-level abstract compiled-code representations. The process begins with preprocessing and compiling the target source code. Subsequently, syntactic features are extracted from the normalized source code, which has been transformed into an AST. Semantic features are then derived from the compiled file using various low-level representations generated by static code analysis tools. In this work, we focus on two specific IRs, namely Baf and Jimple, from the Soot framework, which provides four IRs for Java source code. The extracted syntactic and semantic features are combined to form a comprehensive representation of the source code. This integrated representation serves as the basis for training different machine learning classifiers, which will be used to detect code clones.

The contributions of this work are summarized as follows:

- New integrated code representation: This work introduces a novel integrated code representation that combines high-level and low-level abstract compiled code representations for Java source code. To our knowledge, this is the first attempt to utilize features from both Baf and Jimple IRs to represent the syntactic and semantic features of Java source code fragments.

- Generalized machine learning model: This work proposes a generalized machine learning model designed for the identification of syntactic and semantic clones in Java source code, utilizing the proposed code representation.

- Extensive experiments were performed using the dataset from BigCloneBench [16], a benchmark dataset containing well-labeled clone pairs. The objective was to assess the effectiveness of the proposed technique and compare it with state-of-the-art clone detection techniques in terms of recall and F1 score.

The study is structured as follows: Section 2 provides an outline of the background. Related research is presented in Section 3, while Section 4 discusses the research methodology. Section 5 presents our experimental results on a real-world dataset, followed by a discussion of the results in Section 6. Section 7 addresses potential threats to validity. Finally, Section 8 concludes the article and delineates avenues for future work.

## 2 Background

Code clone detection methodologies typically encompass three fundamental phases: (1) Pre-processing of code, which involves removing extraneous elements such as header files and comments; (2) Transformation of source code into an intermediary representation (such as AST, sequence of tokens, or Program Dependency Graphs (PDG)); and (3) Comparison of code similarity, wherein the similarity between code fragments is calculated, facilitating the detection of code clones if this similarity surpasses a predetermined threshold. This section introduces the foundational concepts underlying the techniques used in this work.

### 2.1 Abstract Syntax Tree (AST)

An AST is a tree-like representation of source code, defining its abstract syntactic structure [17]. Unlike conventional textual source code, ASTs offer an abstraction that omits finer details such as punctuation and delimiters. Within an AST, nodes align with programming constructs or symbols present in the source code. Researchers leverage this tree structure to precisely identify programming constructs, including assignment statements, declaration statements, and various operations. This capability facilitates tasks like optimization, analysis, and code modification. For a visual representation, Fig 1 shows the syntactic structure of the f () function in the form of an AST.

After generating a tree representation for code fragments, researchers have several options. They can choose to detect similarities between corresponding ASTs or their sub-trees. Alternatively, they can convert them into a sequence of tokens, denoted as [token-1, token-2, . . ., token-n], or represent the frequency of each programming construct in the AST (tokens) by traversing the AST. The comparison of ASTs from two code fragments yields a distance value that quantifies their similarity [18].

In this study, JavaParser tool [19] was used to generate and process ASTs of code fragments. JavaParser [20] is a Java library designed to facilitate the parsing, manipulation, and analysis of Java source code. By representing code as ASTs, developers can interact with Java code programmatically, gaining insight into its syntax and structure. JavaParser streamlines tasks such as parsing Java source code, navigating its structure, and programmatically modifying it.

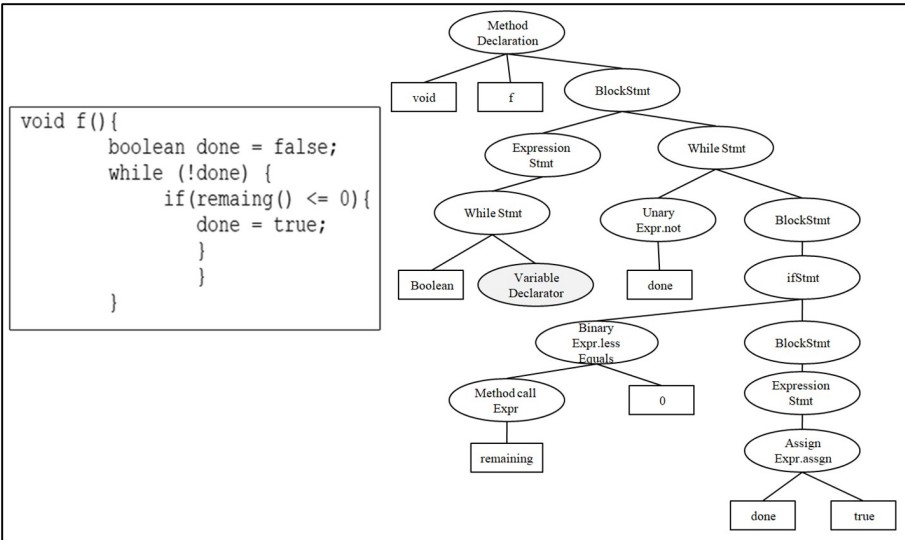

**Fig 1. Example of AST extracted from f() function.**

This library finds widespread use in software development tools, static code analysis tools, and frameworks that necessitate the analysis or manipulation of Java code.

## 2.2 Soot framework

The Soot framework [21] is an open-source program designed for the optimization, analysis, transformation, and visualization of Java and Android applications. One of its core strengths lies in its capacity to convert programs into different IRs. Each of these IRs offers distinct levels of abstraction, providing multiple advantages for source code analysis. Soot offers four IRs for the analysis and transformation of Java code: Baf, Jimple, Shimple, and Grimp. However, this work primarily focuses on the Baf and Jimple IRs. Further details are available in [22].

**2.2.1 Baf intermediate representation.** Baf is a stack-based bytecode representation, similar to Java bytecode but simplified. It abstracts the constant pool and consolidates type-dependent variations of instructions into single instructions. For instance, in Fig 2, two functions perform equivalent operations: one function adds two integers, while the other adds two floating-point numbers. To identify the similarity between these two functions at the low-level representation, a compiler, such as Javac, is employed to process both functions and retrieve the corresponding bytecode instructions using a Javap for subsequent comparison.

```
public static int Integer_Sum()
{
        int x=80;
        x= x+20;

        return x;
}
```

```
public static float Float_Sum()
{
        float f - 80;
        f=f+20;

        return f;
}
```

**Fig 2. Two functions for summation using integer and float data types.**

```
public static int Integer_Sum();
        Code:
            0: bipush      80
            2: istore_0
            3: iload_0
            4: bipush      20
            6: iadd
            7: istore_0
            8: iload_0
            9: ireturn
```

```
public static float Float_Sum();
        Code:
            0: ldc        #3 // float 80.0f
            2: fstore_0
            3: fload_0
            4: ldc        #4 // float 20.0f
            6: fadd
            7: fstore_0
            8: fload_0
            9: freturn
```

**Fig 3. Two functions for summation using integer and float data types.**

As illustrated in Fig 3, it is apparent that the bytecode instructions for each of the two functions display a significant dissimilarity. This divergence arises from the inherent diversity of Java bytecode instructions, each designed to operate with specific data types, such as iadd and fadd. Consequently, the task of identifying similarities between code fragments at the low-level representation becomes notably challenging. In fact, Java bytecode encompasses an extensive set of over 250 distinct instructions [23]. The Baf IR contains approximately 60 instructions designed to represent code fragments in an abstract bytecode format. As illustrated in Fig 4, it is evident that a significant level of resemblance can be observed among the Baf IR instructions employed in the two functions presented in Fig 4.

Consequently, using the capability of Baf IR to represent bytecode abstractly will improve the effectiveness of code clone detection techniques in detecting more difficult clones like semantic clones.

**2.2.2 Jimple intermediate representation.** In the Soot framework, the primary intermediate representation is called Jimple. Jimple serves as a typed 3-address code, containing only 15 statements to represent the source code [24, 25]. It acts as an intermediary layer situated above the stack-based Java bytecode, effectively replacing it. Within the Soot framework, Jimple is employed for generating optimized IR, a process that involves the elimination of dead code, unused local variables, and common sub-expressions. Furthermore, during the transformation to Jimple, expressions are linearized to ensure that statements reference, at most, three local variables or constants. To illustrate this process, refer to Fig 5, which presents a code fragment containing dead code and unused variables, along with its corresponding optimized

```
public static int Integer_Sum()
{
        push 80;
        push 20;
        add;

        return;
}
```

```
public static float Float_Sum()
{
        push 80.0F;
        push 20.0F;
        add;

        return;
}
```

**Fig 4. Baf IR for the *Integer_Sum*() and *Float_Sum*() functions.**

```
public static float validate(){
    float sum = 0;
    // unused variable
    double r = 8.0;
    // start dead code section
    int x = 9;
    if(x < 9){
        System.out.println("Dead code");
        System.out.println("Dead code");
        System.out.println("Dead code");
                }
    //end dead code section
    sum = sum + 5;
    return sum;
    }
```

```
public static float validate()
    {
        float f1;
        goto lable1;

    labels:
     f1 = 0.0F + 5.0F;

     return f1;
    }
```

**Fig 5. Code fragment *validate*() and its optimized Jimple IR.**

Jimple IR. As demonstrated in the Jimple IR, the dead code and unused variables have been successfully eliminated.

Furthermore, in Jimple IR, various programming constructs, including do-while, for, and while loops are mapped to equivalent Jimple statements, such as if and goto statements, as shown in Fig 6. These optimizations lead to a substantial reduction in the number of operations needed for an efficient representation of Java bytecode, thereby enhancing the similarity between code fragments.

Utilizing optimized Jimple IR for code fragments offers substantial advantages by eliminating extraneous elements while retaining the fundamental operations that convey the semantic meaning of the code fragment. This representation diminishes the effectiveness of obfuscation techniques, such as those that alter the syntactic structure while preserving the original code's

```
public static float validate(){
    float c = 90;
    float f = 0;
    for(int i=0;i<c;i++){
        f = f + i; }
    return f;
    }
```

```
public static float validate(){
    float c = 90;
    float f = 0;
    int i = 0;
    while(i<c){
        f = f + i;
        i++; }
    return f; }
```

```
public static float validate(){
    float $f1, $f2, f3;
    byte $b0;
    int i1;
    f3 = 0.0F;
    i1 =0;
    label:
    $f1 = (float) i1;
    $b0 = $f1 cmpg 90.0F;
    if $b0 >= 0 goto label2;
    $f2 = (float) i1;
    f3 = f3 + $f2;
    i1 = i1 + 1;
    goto label1;
    label2:
     return f3;
    }
```

```
public static float validate(){
    float $f1, $f2, f3;
    byte $b0;
    int i1;
    f3 = 0.0F;
    i1 =0;
    label:
    $f1 = (float) i1;
    $b0 = $f1 cmpg 90.0F;
    if $b0 >= 0 goto label2;
    $f2 = (float) i1;
    f3 = f3 + $f2;
    i1 = i1 + 1;
    goto label1;
    label2:
     return f3;
    }
```

**Fig 6. Code fragments with different loop statements and their Jimple IR.**

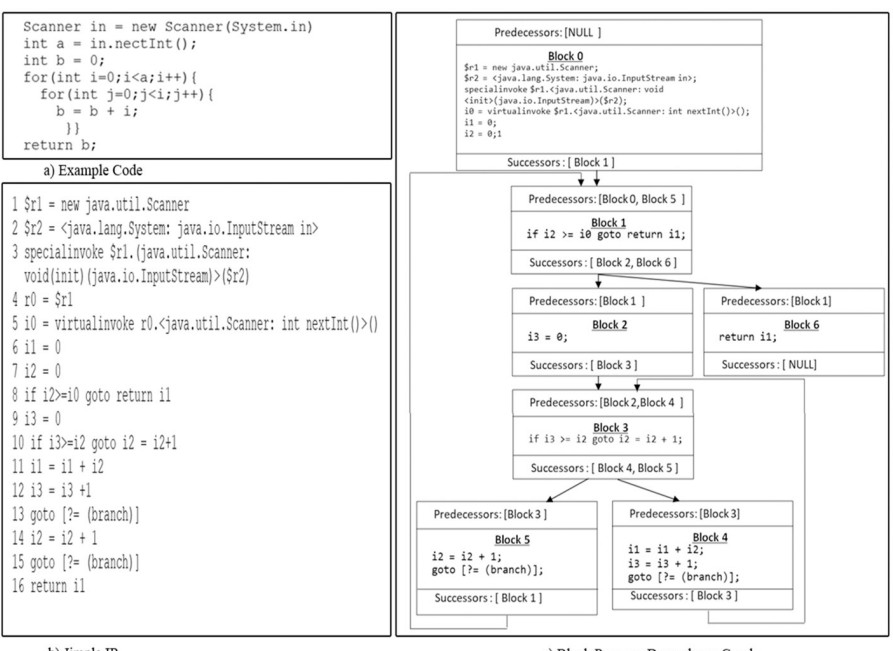

**Fig 7. Jimple block program dependence graph example.**

semantics and functionality (e.g., metamorphism and polymorphism [26]). Furthermore, it facilitates the detection of duplicated or camouflaged code fragments by research tools.

**2.2.3 Jimple Program Dependency Graph (PDG).**   A PDG is a labeled, directed graph used to depict control and data dependencies among elements within a program. In the PDG, nodes represent assignment statements and control predicates, while edges signify control and data dependencies, as well as program component execution order. Data dependencies capture relevant data flow and control dependencies represent essential control flow relationships. PDGs are valuable across various software engineering and reengineering tasks. To obtain structural insights from source code, analysis can be performed on the source code itself, binary code, or any intermediate representation (e.g., Java bytecode, Jimple IR, LLVM bitcode). This work focuses specifically on block PDG created from the Jimple intermediate representation using the Soot framework.

Fig 7 depicts a code example alongside its corresponding Jimple IR and their block PDG. In this block PDG, each rectangle represents a basic block, within which multiple intermediate instructions may exist. Both data dependencies and control dependencies can be extracted for each basic block. For instance, in basic block 1, a conditional branch dictates the execution flow between basic blocks 2 and 6.

## 3 Related work

Over the past decade, many approaches for detecting code clones have been proposed. However, the majority rely on high-level source code representations, with only a limited number using low-level source code representations.

It's widely recognized that the text-based, token-based, syntax-based (AST), semantic-based (PDG), and metrics-based clone detection techniques all rely on high-level source code representations as their foundation. Cordy and Roy [27] introduced NiCad, a text-based tool

that effectively detects Type-III clones. This is achieved by normalizing source code using specific transformation rules and employing the longest common subsequence matching algorithm to identify similarities and reach a final decision. SourcererCC [28], introduced by Saini et al., is a token-based technique that generates tokens from source code. It employs optimized token indexing and filtering heuristics to detect code clones, both within the same project and different across projects. Tiancheng et al. [29] introduced an innovative technique aimed at simplifying the complex structure of the AST. This approach accelerates the process of detecting and locating code clone fragments. Furthermore, Kamalpriya and Singh [30] introduced a detection approach that employs the ASM technique to improve PDG-based code clone detection. Their method derives a novel similarity relationship from the findings obtained through existing PDG-based techniques. The ASM technique is employed to generate these new approximate clone relationships. Hua et al. [31] introduced a deep-learning-based code clone detection approach called FCCA. This method employs a hybrid code representation, including sequences of tokens, abstract syntax trees (AST), and control flow graphs (CFG), alongside an attention mechanism to accurately identify complex functional code clones in real-world codebases. In a similar vein, Fang et al. [13] proposed a novel approach to functional code clone detection. Their method introduces a joint code representation comprising AST and CFG and adopts fusion embedding techniques and supervised deep learning to achieve superior performance in clone detection. Basit and Jarzabek [32] and Ali et al. [33] presented code clone detection as a data mining problem and used data mining techniques to detect code clones at various levels. Heba and El-Hafeez [34, 35] applied association analysis in data mining to tackle redundancy and select relevant features for text classification, a methodology that could be extended to code clone detection.

However, certain approaches concentrate on utilizing low-level source code representations, such as bytecode or assembly instructions or other intermediate representations, as the basis for clone detection. Andre et al. [36] used paths from the dominator tree within the control flow graph of Java bytecode to represent code fragments. Salim et al. [24] suggested a clone detection approach by converting Java source code into Jimple IR using the Soot framework. Then they used text-based detection techniques to locate clones at this abstraction level. Roy et al. [37] introduced SeByt, a model for detecting bytecode clones. They segmented the bytecode into three dimensions, comprising instructions, method calls, and types, and performed semantic searches on the bytecode ontology to match the bytecode content. Caldeira et al. [38] enhanced clone detection by combining the simplicity of the text-based techniques with the abstractions granted by intermediate representations. They converted the C source code into an intermediate representation using the LLVM virtual assembly language. After that, they executed NiCad on the generated IR. Yu et al. [39] used the Smith-Waterman algorithm to align sequences of Java bytecode in order to identify clones in Java code.

Oleksii et al. [40] have demonstrated that code clone detection techniques yield significantly different results when applied to the same source code with different code representation levels, namely high-level and low-level code representations. Consequently, different techniques have emerged to enhance clone detection efficiency by integrating code representations from different levels. Sheneamer et al. [41] introduced a new framework for detecting semantic code clones and code obfuscation. They enhanced the detection using a hybrid code representation derived from bytecode dependency graph (BDG), AST, and PDG features. White et al. [8] propose a technique that uses recurrent neural networks to learn source code representation by integrating features extracted from the CFG, sequence of identifier tokens, and sequence of bytecode instructions. Moreover, Tufano et. al. [42] uses four diverse code representations (i.e., sequence of identifier tokens, AST, bytecode, and CFG) to assess the similarity between pairs of code.

**Table 1. Code representation techniques used in the literature.**

| Technique | Type | Code Representation | | | | | |
|---|---|---|---|---|---|---|---|
| | | Text | Token | Tree | Graph | Bytecode | IR |
| Roy and Cordy | H | ✓ | | | | | |
| SourcererCC | H | | ✓ | | | | |
| Tiancheng et al | H | | | ✓ | | | |
| Kamalpriya and Singh | H | | | | ✓ | | |
| Hua et al. | H | | ✓ | ✓ | ✓ | | |
| Fang et al. | H | | | ✓ | ✓ | | |
| Basit and Jarzabek | H | | ✓ | | | | |
| Ali et al. | H | | ✓ | | | | |
| Andre et al. | H | | | ✓ | ✓ | | |
| Salim et al. | L | | | | | | ✓ |
| Roy et al. | L | | | | | ✓ | |
| Caldeira et al. | L | | | | | | ✓ |
| Yu et al. | L | | | | | ✓ | |
| White et al. | L | | | | ✓ | | |
| Sheneamer et al. | H/L | | | | ✓ | ✓ | |
| Tufano et. al. | H/L | | ✓ | ✓ | ✓ | ✓ | |

L : indicates the low level representation; H : indicates the High-level representation; IR : indicates the intermediate representation

The proposed work aligns with the concept of integrating the syntax representation of the AST and the semantics representation of the compiled code and program dependency graph. Table 1 illustrates the code representation techniques employed in the literature.

## 4 The proposed methodology

An ideal code representation should comprehensively preserve code features. This section describes the methodology used for the comprehensive representation and extraction of both syntactic and semantic features from source code to detect syntactic and semantic clones. The methodology encompasses three stage: pre-processing, feature extraction, and machine learning (ML) training. Fig 8 provides a more detailed overview of this methodology.

### 4.1 Stage 1: Pre-processing stage

In the proposed methodology, as part of the preprocessing stage, all files within the source code corpus undergo normalization and compilation utilizing a compilation tool such as the Stubber tool [43, 44]. This stage involves several normalization operations, including the removal of unnecessary white spaces, the exclusion of comments, the substitution of variable names and function names with generic placeholders to mitigate disparities in naming conventions, and the substitution of numeric literals, string literals, and other constants with placeholders to generalize specific values. After normalizing and compiling files in this stage, they proceed to the subsequent processing steps.

### 4.2 Stage 2: Features extraction stage

The subsequent stage, known as the features extraction stage, comprises two sub-stages, as shown in Fig 8. The first sub-stage entails extracting syntactic features, while the second focuses on extracting semantic features. This process initiates with the reception of normalized

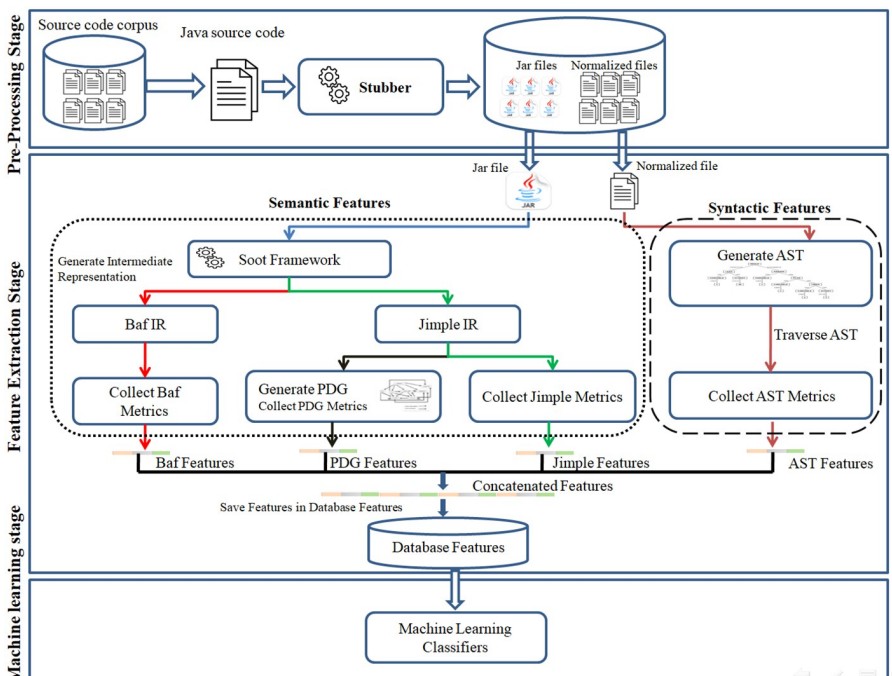

**Fig 8. The proposed work's architecture workflow.**

Java source code and its associated compiled JAR file. Subsequently, it extracts code fragments (methods) from these files and analyzes their corresponding syntactic and semantic features, as illustrated in Fig 9 and further explained in the subsequent subsections.

**4.2.1 Extraction of syntactic features (High-level features).** To capture the abstract syntactic structure of the source code, the normalized Java source file is processed in this sub-stage. It is first converted into an AST using the JavaParser [20] tool to extract all methods in the file. Subsequently, each method is converted into an AST using the same tool, and the AST is traversed to extract method syntactic features. The algorithm for syntactic feature extraction, denoted as Algorithm 1, describes the procedure for extracting syntactic features.

**Algorithm 1**: Extract syntactic features

```
Input: SF: Java source file
Output: ASTFeaturesMap: a Map for AST features
1  ASTFeaturesMap ← ∅; MethodList ← ∅;
   // Convert Java source file into AST
2  Cu ← StaticJavaParser.parse(SF);
   // Retrieve the list of methods in the AST
3  MethodList ← travserse(Cu);
   // Method-level AST features extraction
4  foreach m in MethodList do
     // Retrieve method body
5    Methodbody ← RetriveMethodBody(m, cu);
6    IdentSeq ← ∅;
     // Convert method into AST
7    MAST ← StaticJavaParser.parse(Methodbody);
     // Traverse method AST in pre-order manner
8    IdentSeq ← travserse(MAST);
     // Put method's extracted features in Map
```

```
9    ASTFeaturesMap.put(m, IdentSeq);
10 end foreach
11 return ASTFeaturesMap;
```

This procedure consists of six distinct steps, which are detailed below.

1. Transform the normalized Java source file into an AST using the JavaParser tool [20] (Algorithm 1: Line 2).

2. Retrieve a list of methods contained within the AST (Algorithm 1: Line 3).

3. For each method in the method list, the method's body is extracted (Algorithm 1: Line 5).

4. Transform the method body into an AST using the JavaParser tool (Algorithm 1: Line 7).

5. Iterate through each non-terminal node in the AST of the method in a preorder manner and extract a sequence of programming constructs nodes [token1, token2,. . ., tokenn] (node identifiers). (Algorithm 1: Line 8).

6. Store the extracted sequence of AST nodes' identifiers in a map, along with its corresponding method name. (Algorithm 1: Line 9).

7. Repeat steps 3 to 6 until all the methods have been scanned.

The primary output of the syntactic feature extraction algorithm is a map that contains method names paired with their corresponding sequence of tokens (AST nodes' identifiers). A list of non-terminal nodes investigated in this work is presented in Table 2. For further details, please refer to the S1 File.

**4.2.2 Extraction of semantic features (Abstract compiled-code features).** Higher abstraction enhances a feature representation's capacity to measure source code semantics or meaning. This occurs because greater abstraction allows capturing a broader range of semantic code information [45]. Unlike syntactic features, semantic features represent source code at a

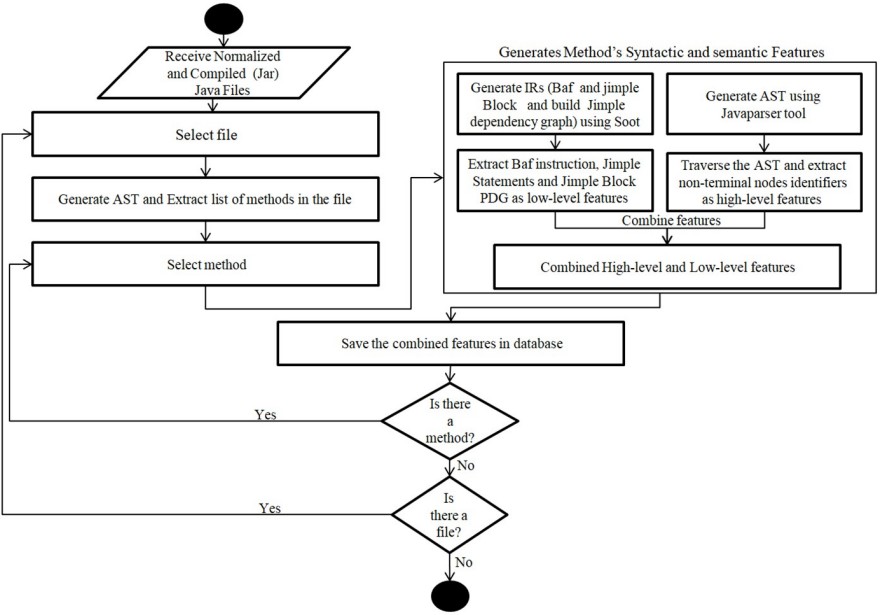

**Fig 9. The process of extracting syntactic and semantic features.**

**Table 2. Selected AST non-terminal nodes.**

| ID | Node | Description |
|----|------|-------------|
| 1 | BlockStmt | Basic block in the method. |
| 2 | ArrayCreations | Arrays declaration in the method. |
| 3 | ObjectCreationExpr | Objects created in the method. |
| 4 | ExpressionStmt | Expression statements in the method. |
| 5 | MethodCallExpr | Methods invoked in the method. |
| 6 | FieldAccessExpr | Field access in the method. (i.e., Person.name) |
| 7 | PrimitiveType | Primitive types in the method. |
| 8 | VariableDeclarationExpr | Variables declared in the method. |
| 9 | WhileStmt | While statements in the method. |
| 10 | ReturnStmt | Return statements in the method. |

higher level of abstraction, capturing logic flows and program execution sequences. To extract the semantics or meaning of a code fragment, the initial step involves parsing it into syntactic or structural components. For each component, its corresponding meaning is determined, and subsequently, the code fragment's overall meaning is constructed by combining these components following relevant rules [41].

In this work, the capabilities of the Soot framework to transform Java programs into different abstract IRs are leveraged to obtain semantic features from the source code. Specifically, two IRs, namely Baf and Jimple, are employed in this research. Furthermore, additional semantic information is extracted by abstractly representing the data and control dependencies within the Jimple IR through the construction of a block PDG. The procedure for extracting Baf IR features is outlined in Algorithm 2.

**Algorithm 2**: Extract Baf Features

```
Input: JClassfile,MethodList: compiled Java file (.classfile) and
List of methods in the target file
Output: BafFeatureMap: Method names and its Baf features
1  BafFeatureMap ← ∅;
2  ML ← ∅;
   // Retrieve methods list from .classfile
3  ML ← Soot.Classfile.getMethods(JClassfile);
   // Method-level Baf features extraction
4  foreach m in ML do
5    if (MethodList.contains(m.getName())) then
       // Retrieve method body in Baf IR
6      Bafbody ← Soot.getBafBody(m);
7      LBafSeq ← ∅;
       // Iterate through Baf units and extract sequence of instructions
8      foreach Unit in Bafbody.getUnits() do
9        LBafSeq.put(Unit.toString());
10     end foreach
       // Put method's Baf extracted features in Map
11     BafFeatureMap.put(m.getName(), LBafSeq);
12   end if
13 end foreach
14 return BafFeatureMap;
```

This algorithm initiates by receiving two parameters: the .classfile and a list of target methods. It then proceeds through the following steps:

1. Retrieve a list of methods contained within the .classfile using the Soot framework (Algorithm 2: Line 3).

2. For each method in the method list, the Soot framework generates the optimized Baf IR of the method (Algorithm 2: Line 6).

3. Iterate through all instructions in the Baf body to extract Baf instructions, forming a sequence of Baf instructions (Algorithm 2: Lines 8 to 10).

4. Store the extracted sequence of Baf instructions in a map, along with its corresponding method name. (Algorithm 2: Line 11).

5. Repeat steps 4 to 13 until all the methods have been scanned.

The main result of this procedure is a map that pairs method names with their respective sequence of Baf instructions. A list of Baf instructions examined in this work is provided in Table 3. For more details, please refer to the S1 File.

Additionally, the Jimple and Block PDG feature extraction algorithm (Algorithm 3), delineates the process of extracting semantic features from the Jimple IR and its corresponding block program dependency graph. The Jimple and Block PDG feature extraction algorithm begins by receiving two parameters: the .classfile and a list of target methods. Then, it proceeds through the following steps:

1. Retrieve a list of methods contained within the .classfile using the Soot framework (Algorithm 3: Line 3).

2. For each method in the method list, the Soot framework generates the optimized Jimple IR of the method (Algorithm 3: Line 8).

3. Iterate through all statement units in the Jimple body to extract a sequence of Jimple statements and form a map containing the frequency of each Jimple statement (Algorithm 3: Lines 10 to 13).

4. Create a block program dependency graph (BPDG) for Jimple IR (Algorithm 3: Line 16).

5. Retrieve block control flow graph (BCFG) features for the Jimple BPDG (Algorithm 3: Line 17).

6. Retrieve BPDG features for the Jimple BPDG (Algorithm 3: Line 18).

**Table 3. Selected Baf instructions.**

| ID | Instruction | Description |
|---|---|---|
| 1 | Load | Load variable from local variable. |
| 2 | Store | Store variable into local variable. |
| 3 | Inc | Increment local variable by constant. |
| 4 | fieldget | Fetch field from object. |
| 5 | New | Create new object. |
| 6 | Ifne | Jump if value1 $\neq$ *value 2* |
| 7 | Ifcmpeq | Branch if and only if Value1 = value 2. |
| 8 | sub | Subtract two varaible. |
| 9 | Add | Add two variables. |
| 10 | Checkcast | Check whether object is of given type. |

7. Store the extracted features from Jimple, Jimple BPDG, and BCFG in a map, along with their corresponding method name. (Algorithm 3: Lines 17 to 21).

8. Repeat steps 4 to 22 until all the methods have been scanned.

**Algorithm 3**: Extract Abstract Jimple and PDG Features

```
Input: JClassfile, MethodList: // .classfile and List of methods in
the target file
Output: JimpleFeatureMap: // method names and its Jimple features
1  JimpleFeatureMap ← ∅;
2  MList ← ∅;
   // Retrieve methods list from .classfile
3  MList ← Soot.Classfile.getMethods(JClassfile);
   // Method-level Jimple features extraction
4  foreach m in MList do
5    if (MethodList.contains(m.getName())) then
6      FturLst ← {JimpleFeatures};
7      JimpleLstMap ← ∅;
       // Retrieve method body in Jimple IR
8      Jimplebody ← (JimpleBody)m.retrieveActiveBody();
       // Iterate through Jimple units and extract the frequency of each
       statements and statement sequence
9      JimpleSeqStmt ← ∅;
10     foreach u in Jimplebody.getUnits() do
11       FturLst.put(FturLst.getKey(u), FturLst.getValue(u) + 1);
12       JimpleSeqStmt ← JimpleSeqStmt + u;
13     end foreach
       // Use Soot to create Block PDG for Jimple IR
14     JBlockGraph ← Soot.EnhancedBlockGraph(Jimplebody);
       // Retrieve Block CFG features for Jimple block graph
15     BCFGfeatures ← getBCFGFeatures(JBlockGraph);
       // Retrieve Block PDG features for Jimple block graph
16     BPDGfeatures ← getPDGFeatures(JBlockGraph);
       // Add extracted features into map features
17     JimpleLstMap.put(JimpleFreq, FturLst);
18     JimpleLstMap.put(JimpleSeqStmt, JimpleSeqStmt);
19     JimpleLstMap.put(bcfg, BPDGfeatures);
20     JimpleLstMap.put(bpdg, BPDGfeatures);
       // Add extracted features in map feature into the general Map
21     JimpleFeatureMap.put(m.getname(), JimpleLstMap);
22   end if
23 end foreach
24 return JimpleFeatureMap;
```

The main output of the Jimple and Block PDG feature extraction algorithm is a map that associates method names with their respective maps, which contain information about Jimple statement frequency, the sequence of Jimple statements, and Jimple Block PDG features. Table 4 presents a list of the Jimple and block PDG features. For more detailed information, please consult the S1 File.

After extracting syntactic features from the source code using Algorithm 1 and obtaining semantic features through Algorithms 2 and 3 for each method within the target Java source code, the extracted features for each method are subsequently combined and then stored within a database. The combined feature extraction algorithm (Algorithm 4), consists of the following sequential steps:

1. Receive two inputs: the target Java source code file and its corresponding .classfile.

**Table 4. Selected Jimple and Block PDG features.**

| ID | Statement | Description |
|----|-----------|-------------|
| 1 | IfStmt | Represent conditional jump. |
| 2 | GotoStmt | Represent unconditional jump. |
| 3 | InvokeStmt | invokeStmt represents an invoke without an assignment to a local. |
| 4 | AssignStmt | Assigning a value to a local, or an immediate static field. |
| 5 | ThrowStmt | Represents the explicit throwing of an exception. |
| 6 | Number of PDG region | |
| 7 | Number of strong regions in PDG | |
| 8 | Number of weak regions in PDG | |
| 9 | Number of dependency Edges in PDG | |
| 10 | Number of control flow edges in PDG | |
| 11 | Number of dependency-back edges in PDG | |
| 12 | Number of dependency edges between Region node and Region node in PDG | |

2. Invoke the syntactic feature extraction algorithm (Algorithm 1 and supply the target Java file as input to retrieve a map containing all the methods in the target file, along with their corresponding sequence of AST nodes identifiers, (Algorithm 4 Line 4).

3. Invoke the extracting Baf IR features algorithm (Algorithm 2 with two parameters: the . classfile of the target Java file and the list of methods contained therein. This step generates a map containing all methods in the compiled file and their corresponding sequence of Baf instructions (Algorithm 4: Line 7).

4. Invoke the Jimple and BPDG feature extraction algorithm (Algorithm 3 and pass two parameters: the .classfile of the target Java file and the list of methods within that file. This operation produces a map containing all the methods in the compiled file, along with their corresponding sequence of Jimple statements, the frequency of each Jimple statement, and BPDG features (Algorithm 4: Line 9).

5. For each method in the method list, obtain the sequence of AST nodes' identifiers, sequence of Baf instructions, sequence of Jimple statements, frequency of each Jimple statement, and Jimple BPDG features. Combine these elements into a single sequence or vector and store it in the features data storage. (Algorithm 4: Lines 10 to 13).

**Algorithm 4**: Combined Extracted Features

```
   Input: JClassf, SF; // Java source file and .classfile
   Output: DataStorage Features;
 1 combinedF ← ∅; // Combined features Map
 2 MList ← ∅; // Method List
 3 // Invoke Algorithm 1: To extract AST features
 4 ASTFMap ← ExtractSyntacticFeatures(SF);
 5 MethodList ← ExtractSyntacticFeatures(SF).MethodList;
 6 BafFMap ← ∅;
   // Invoke Algorithm 2: To extract Baf IR features
 7 BafFMap ← ExtractBafFeatures(Jclassf, MList);
 8 JimpleFMap ← ∅;
   // Invoke Algorithm 3: To extract Jimple IR features and Jimple BPDG
   features
 9 JimpleFMap ← ExtractJimplePDGFeatures(Jclassf, MList);
10 foreach m in MList do
```

```
      // Combine the extracted features for each method and store it in
   DataStorge
11    combinedF ← ASTFMap.get(m) + "," + BafFMap.get(m) + "," + Jim-
   pleFMap.get(m);
12    InsertFeatureIntoDataBase(m, combinedF);
13 end foreach
```

After applying all of these steps on the entire corpus and extracting features, which are then stored in the data storage, this work involves creating pairs of instances from the extracted features. If two original instances are represented by feature vectors $X = (x_1, x_2, \ldots, x_n)$ and $Y = (y_1, y_2, \ldots, y_n)$, a pair instance $Z = (X, Y)$ is represented as a vector using one of the following combination techniques:

- Linear Combination.

$$Z(X, Y) = (x_1, x_2, ..., x_n, y_1, y_2, ..., y_n) \tag{1}$$

- Distance Combination

$$Z(X, Y) = (|x_1 - y_1|, |x_2 - y_2|, ..., |x_n - y_n|) \tag{2}$$

- Multiplicative Combination.

$$Z(X, Y) = (x_1 * y_1, x_2 * y_2, ..., x_n * y_n) \tag{3}$$

## 4.3 Stage 3: Machine learning stage

In machine learning, classifiers categorize data points into specific classes using various techniques such as linear and non-linear models, probabilistic and non-probabilistic methodologies, decision trees, and more. In this phase, fifteen classification algorithms are selected, including ensemble and individual techniques, to evaluate their effectiveness in detecting semantic clones using the proposed features. The following subsection describes the classifiers used:

**4.3.1 Ensemble classifiers.** Ensemble classifiers integrate base classifiers to improve prediction performance by reducing the misclassification rate of a weak classifier through the aggregating of multiple classifiers [46]. The key strategies employed in ensemble classifiers include bootstrap aggregation (bagging) [47] and boosting [48].

- **Boosting Classifiers:** CatBoost [49] employs gradient boosting on decision trees, with strategies to prevent overfitting. XgBoost [50], an extreme gradient boosting implementation, corrects errors from preceding models for final predictions. LightGBM [51] follows the gradient-boosting decision tree (GBDT) method for data modelling, consolidating weak learners into a robust model. LogitBoost [52] progressively enhances base learners' performance by addressing misclassified instances [53].

- **Boosting Classifiers:** Random Committees [54] combine predictions from individually trained models. In contrast, the Random forest [55] aggregates multiple decision trees, each created by sampling from the input vector. Rotation Forest [47] builds on the Random Forest concept but trains decision trees independently on a rotated feature space. Bagging [56]

constructs multiple independent classifiers from training instances and aggregates their predictions. However, the Random Subspace technique [57] involves training models on diverse random feature subsets.

**4.3.2 Individual classifiers.** J48 [58], a Java implementation of C4.5 algorithm, normalize bias using the gain ratio in decision tree classification. Naive Bayes [59] applies Bayes' theorem, assuming feature independence. Linear Discriminant Analysis (LDA) [60] is widely used for dimensionality reduction and classification. SVMs [61, 62] optimize decision boundaries to maximize margin between classes. Logistic Regression [63] constructs statistical models describing relationships between dependent and independent variables. FeedForward Neural Network (FFNN) [64] processes data linearly through hidden layers.

# 5 Experimental evaluation

In this section, the dataset used in this work is discussed, and various experiments are conducted to evaluate the effectiveness of the proposed code representation in detecting code clones. These experiments carried out using a workstation equipped with a 2.2 GHz Intel Core i7 CPU, 32GB of (DDR3L 1333/1600 memory), and a 1-Terabyte Solid State Drive.

## 5.1 Dataset

Experiments in this study were carried out on a real-world dataset: BigCloneBench, a popular benchmark for assessing Java code clone detection systems, introduced by Svajlenko et al. [16]. It contains 55,499 Java source files from 24,557 distinct open-source projects, collected through the mining process of IJaDataset-2.0 [65]. Domain experts meticulously labeled this dataset, distinguishing between clones and non-clones based on specific functionalities, without relying on clone detection tools. The current version of this benchmark includes over 8.5 million labelled true clone pairs and more than 260,000 labelled false clone pairs across 43 functionalities, categorized into Type-I, Type-II, Type-III, and Type-IV [16]. Clones falling between Type-III and Type-IV are classified based on their syntactic similarity into four classes: VST3, ST3, MT3, and WT34 clones [66].

Since the proposed approach uses High-level and low-level abstract compiled code representations, the conducted experiments exclusively used the compiled version of BigCloneBench [44]. It's important to note that this compiled version shows slight discrepancies in clone counts compared to the figures outlined in the original BigCloneBench paper [16]. This discrepancy can be attributed to the Stubbler [43] tool's capability to compile only approximately 95% of all Java files within the BigCloneBench dataset, preserving 92.5% of all clones in the dataset.

From the compiled BigCloneBench benchmark, 27 functionalities were selected to construct the dataset. Java files and their associated JAR files for these functionalities were processed using the proposed methodology, generating syntactic features (AST) and semantic features (Baf and Jimple IRs, Jimple PDG). A total of 41,865 Java files were processed, containing 515,654 functions, resulting in 5,968,621 function-level clone pairs and 147,390 non-clone pairs. All clone types meeting or exceeding size criteria (6 lines or 50 tokens) were considered, aligning with benchmarking standards [66, 67]. Dataset details are provided in Table 5. For further details, please refer to the S2 File.

**Table 5. Details of dataset information.**

| Clone Type | Pair Sample | Percentage |
|---|---|---|
| Type-I&II | 23,860 | 0.4% |
| VST3 | 3835 | 0.064% |
| ST3 | 11866 | 0.199% |
| MT3 | 62654 | 1.05% |
| WT3/4 | 5866406 | 98.287% |
| Total | 5968621 | |

## 5.2 Evaluation

The effectiveness of the proposed code representation in identifying code clones, particularly semantic clones, is evaluated through a series of experiments. Various combinations of feature fusion techniques and different feature types and sizes are explored to demonstrate their effectiveness in achieving exceptional detection accuracy. Furthermore, the proposed technique is compared with widely used approaches for clone detection. Due to space constraints, only the best-performing classifiers from most experiments are reported in this section. For additional results, readers are encouraged to refer to the S4 File.

**5.2.1 Evaluation of the performance of various classifiers.** In this experiment, an evaluation was conducted on the performance of three feature concatenation techniques and classifiers. A total of 60,000 pair samples were randomly selected for both Type-I & II, VST3, ST3, MT3, WT3/4 clones, and false positives from the compiled BigCloneBench dataset as outlined in section 5.1. To address the issue of imbalanced data, the standard Synthetic Minority Oversampling Technique (SMOTE [68]) was used to balance the samples for VST3 clone pairs. The training and testing of all candidate classifiers were performed using 10-fold Stratified cross-validation, ensuring consistent class ratios within each fold. Fig 10 depicts a performance comparison in accuracy among all fifteen classifiers utilizing multiplicative, distance, and linear combination approaches on the selected dataset.

The experimental findings highlight the effectiveness of ensemble techniques, which include bagging and boosting methods such as Rotation Forest, Random Forest, Bagging, Random Committee, XGBoost, LightGBM, and CatBoost. These techniques consistently outperform most standalone classifiers, except for the Feed Forward Neural Network (FFNN). This superiority arises from ensemble methods' ability to combine multiple models, strategically leveraging diverse classifiers' strengths to compensate for individual weaknesses [69]. Additionally, ensembles exhibit enhanced robustness to outliers and noisy data, effectively alleviating model bias and variance [70].

Significantly, Random Forest, Rotation Forest, LightGBM, and Xgboost classifiers demonstrate outstanding accuracy among other ensemble classifiers achieving (95.71%, 91.67%, 94.49%), (96.27%, 92.17%, 95.17%), (96.75%, 93.09%, 95.62%), and (96.73%, 93.76%, 95.8%) in linear, multiplicative, and distance combination respectively. It is noteworthy that the FFNN demonstrates competitive performance with ensemble techniques, achieving (92.22%, 87.8%, and 91.23%) accuracy in linear, multiplicative and distance combinations, demonstrating proficiency in handling complex data relationships and acquiring hierarchical representations [71].

Observations across all classifiers indicate that a linear combination consistently yields superior results compared to distance and multiplicative combinations. The effectiveness of Linear combination lies in its ability to preserve the original feature values without alteration, a characteristic not shared by the other two combination methods [72]. The use of

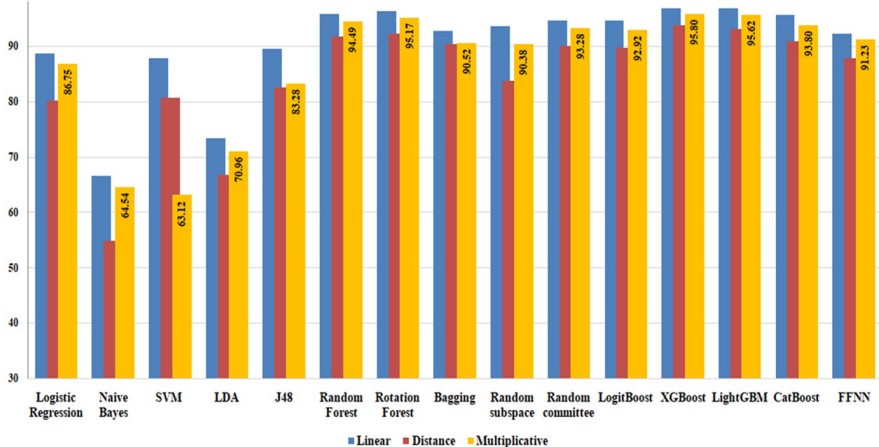

**Fig 10. Performance comparison of fifteen classifiers utilizing various feature combinations on the BigCloneBench dataset.**

multiplicative and distance combinations may potentially hinder classifier performance, as these methods might find certain combinations unsuitable for effective utilization.

**5.2.2 Evaluation of the performance with different dataset sizes and feature types.** Multiple experiments were conducted to evaluate the effectiveness of combining AST, BAF, and Jimple PDG features using three combination methods: Linear, distance, and multiplicative. The datasets employed in these experiments were subsets of BigCloneBench, as specified in section 5.1, comprising 10,000, 20,000, 30,000 and 40,000 paired samples representing both code clones and false positives. Figs 11–13 display the results generated by the highest-performing classifiers in these experiments. Among these top-performing classifiers are two derived from bagging techniques (Rotation Forest and Random Forest), two from boosting techniques (LightGBM and XgBoost), and one individual classifier (Feedforward Neural Network—FFNN).

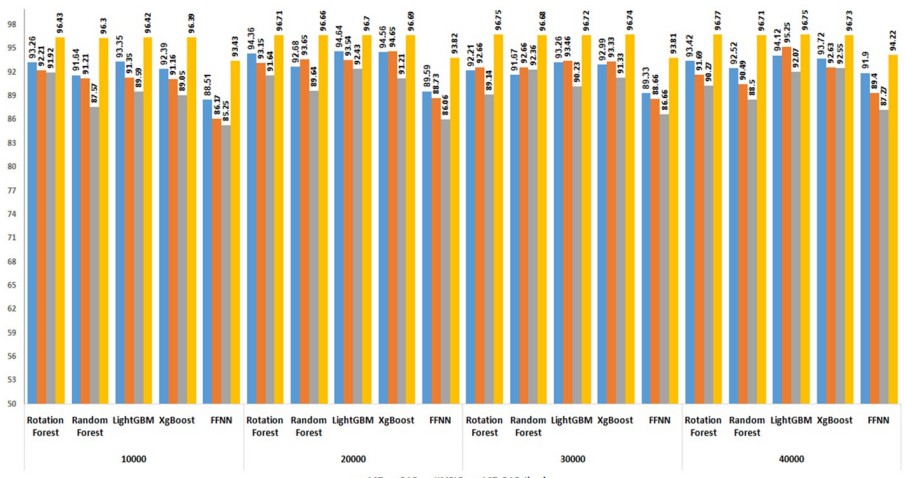

**Fig 11. Performance of top 5 classifiers with linearly combined features on four BigCloneBench datasets.**

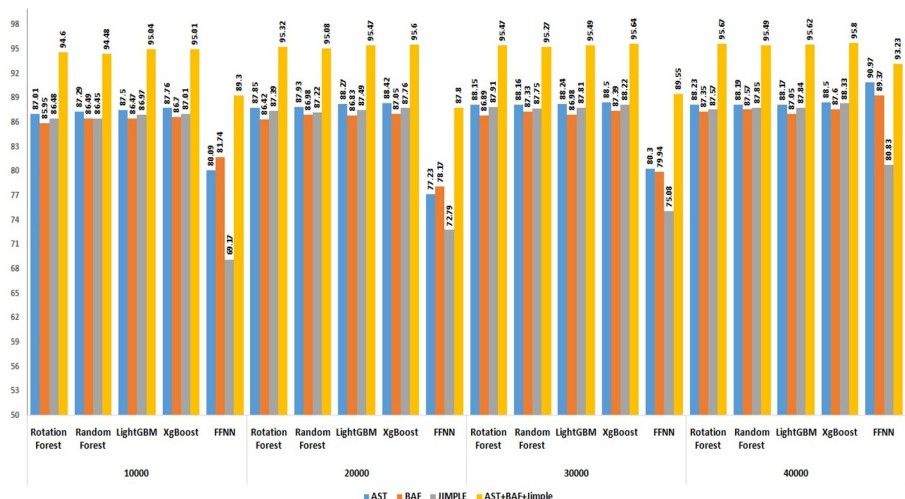

**Fig 12. Performance of top 5 classifiers with multiplicative combined features on four BigCloneBench datasets.**

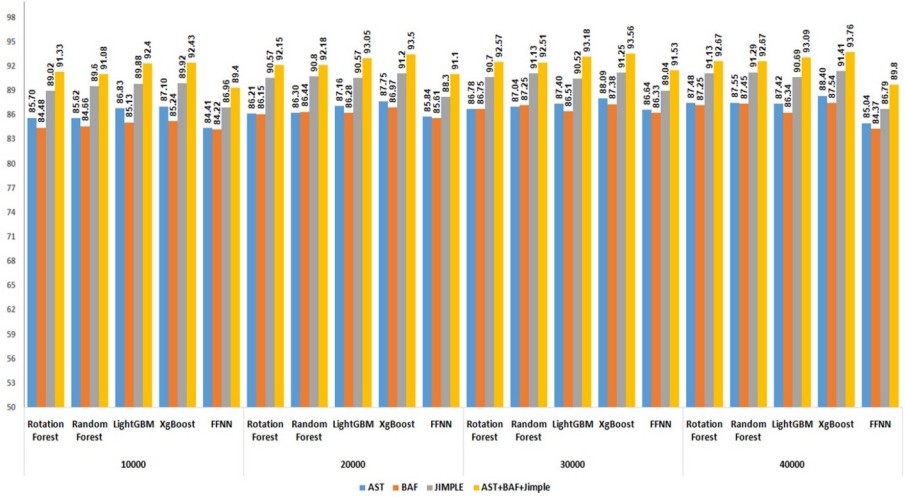

**Fig 13. Performance of top 5 classifiers with distance combined features on four BigCloneBench datasets.**

The integration of AST, BAF, and Jimple PDG features collectively resulted in a significant enhancement in the performance of classifiers for clone detection. In the Linear combination experiments, the average performance disparity between using AST alone and incorporating the combined features was 3.6%. In contrast, the multiplicative combined technique exhibited an average difference of 7.4%, while the distance combined technique showed a difference of 5.4%. Interestingly, it was observed that the classifiers consistently performed well regardless of the dataset size. Additionally, it is noteworthy that a linear combination consistently produced superior results compared with the distance and multiplicative combination methods across all dataset sizes. This finding is consistent with the outcomes of the first experiment results.

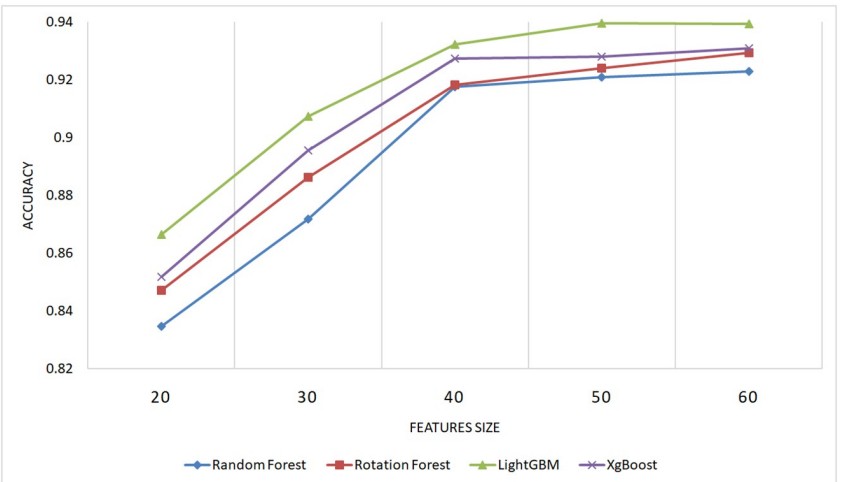

**Fig 14. Performance of XgBoost, LightGBM, Random Forest, and Rotation Forest classifiers with different feature sizes using distance combination approach on the BigCloneBench dataset.**

**5.2.3 Evaluation of the performance with different feature sizes.** To underscore the importance of the proposed features in enhancing performance in code clone detection and to ensure that this improvement is not coincidental, an experiment was conducted involving varying numbers of features, combined by three combination approaches (linear, multiplicative, distance). An Equal number of features was selected from each type (AST, BAF, and Jimple PDG). The dataset utilized in this experiment was obtained from BigCloneBench, specified in section 5.1, comprising 40,000 paired samples representing different types of code clones and false positives.

Figs 14–16 present the outcomes produced by top-performing classifiers, namely Rotation Forest, Random Forest, LightGBM, and XgBoost.

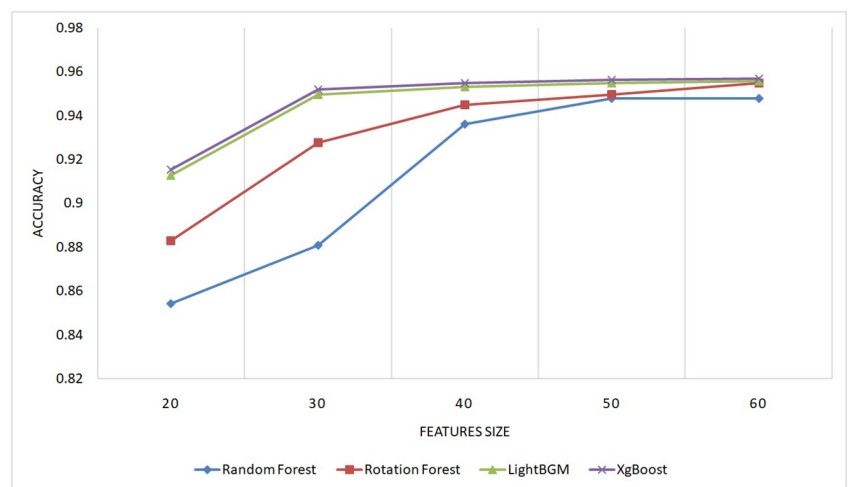

**Fig 15. Performance of XgBoost, LightGBM, Random Forest, and Rotation Forest classifiers with different feature sizes using multiplicative combination approach on the BigCloneBench dataset.**

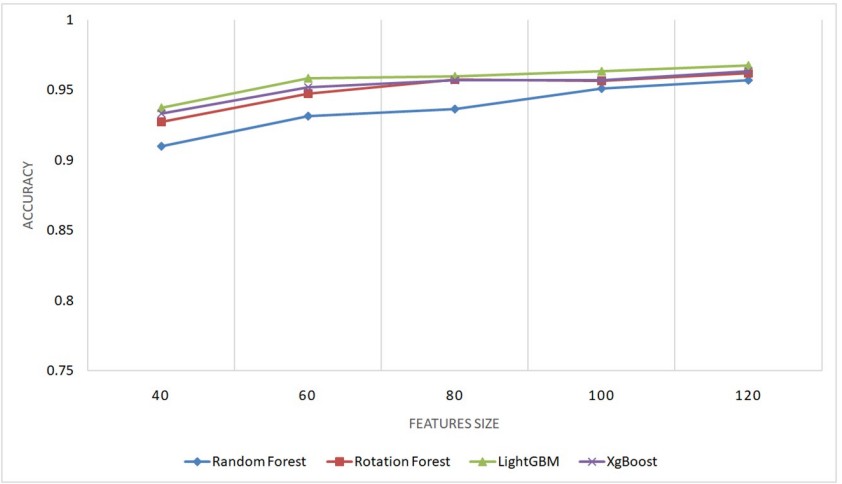

**Fig 16. Performance of XgBoost, LightGBM, Random Forest, and Rotation Forest classifiers with different feature sizes using linear combination approach on the BigCloneBench dataset.**

The ascending learning curve depicted in these figures unequivocally demonstrates that the detection performance improves with the increasing number of features. This observation substantiates the critical role of the proposed features in achieving high-performance detection results.

**5.2.4 Evaluation of the performance in semantic clone detection.** To evaluate the effectiveness of the proposed code representation in detecting semantic clones, particularly those categorized as VST3, ST3, MT3, and WT3/4), five clone detection models were built. Each model employed a distinct classifier. Specifically, the utilized classifiers were Random Forest, Rotation Forest, LightBMG, XgBoost, and FFNN classifiers.

Five experiments were conducted using a dataset containing 20,000 paired samples. Within each clone category (VST3, ST3, MT3, and WT3/4), there were 5000 pairs of samples, randomly selected from a larger dataset sourced from BigCloneBench. These paired samples were amalgamated using three different approaches: linear, multiplicative, and distance. The models underwent training and testing using a 10-fold Stratified cross-validation approach. Table 6 presents the precision, recall, and F1-score for the outcomes of the models in these experiments that utilized paired samples combined using the linear combination approach.

Table 6 reveals that models built with Random Forest, Rotation Forest, and XgBoost classifiers perform well in detecting semantic clones. However, the model constructed using the LightGBM classifier achieves the highest scores across all metrics, including F1-score, precision, and recall, for detecting semantic clones (VST3 F1-score: 97.04%, ST3 F1-score: 96.9%, MT3 F1-score: 96.4%, and WT3/4 F1-score: 92.3%).

Generally, employing our code representation yields commendable results in the identification of semantic clones. This outcome primarily stems from the effectiveness of the proposed code representation. This integration enables the capture of more comprehensive semantic features, thereby enhancing the classifiers' efficiency in identifying a broader range of Type-III and Type-IV clones.

**5.2.5 Performance comparison of different clone detection techniques.** To validate the effectiveness of the proposed technique, a comparative analysis was conducted against selected clone detection techniques that utilized the BigCloneBench dataset. Performance evaluation was assessed using recall and F1-score, widely accepted metrics for classification evaluation.

**Table 6. Results of detected semantic clones using the proposed technique.**

| Classifier | Clone Type | Precision | Recall | F1-score |
|---|---|---|---|---|
| Random Forest | VST3 | 91% | 90% | 90% |
|  | ST3 | 84% | 78% | 81% |
|  | MT3 | 90% | 90% | 90% |
|  | WT3/4 | 79% | 85% | 82% |
| Rotation Forest | VST3 | 91% | 91% | 91% |
|  | ST3 | 85% | 79% | 82% |
|  | MT3 | 91% | 91% | 91% |
|  | WT3/4 | 78.8% | 85% | 82% |
| XgBoost | VST3 | 90.6% | 92% | 91% |
|  | ST3 | 83% | 80% | 82% |
|  | MT3 | 95% | 91% | 93% |
|  | WT3/4 | 81% | 86% | 83% |
| LightBGM | VST3 | 97.9% | 96.2% | 97.04% |
|  | ST3 | 97.018% | 96.8% | 96.9% |
|  | MT3 | 96.5% | 96.3% | 96.4% |
|  | WT3/4 | 95% | 89.7% | 92.3% |
| FFNN | VST3 | 88% | 89% | 89% |
|  | ST3 | 75% | 70% | 72% |
|  | MT3 | 93% | 87% | 90% |
|  | WT3/4 | 71% | 80% | 75% |

The analysis extends beyond overall detection performance; it delves into intricacies across different semantic clone types, specifically VST3, ST3, MT3, and WT3/4. The selected baselines include:

- **Traditional Techniques** encompass a variety of clone detection methods, including Deckard [18], a widely used syntactical-based detector that generates representative vectors for each AST through predefined rules. In contrast, SourcererCC [28] operates as a lexical-based clone detector. Additionally, CCFinder [73] is a popular multilingual token-based clone detector employing a suffix-tree matching algorithm. Nicad [27] employs a straightforward text-based approach for clone detection. iClones [74] uses a hybrid token and tree approach. It creates an AST from the source code, converts it to tokens, and then detects clones using a suffix tree algorithm.

- **Machine learning and deep learning techniques** are exemplified by CCLearner [75], the first token-based detector employing deep learning. CDLH [76] represents a clone detection approach based on deep learning, utilizing ASTs to represent code features. Oreo [7] is a code clone detector that integrates software metrics, information retrieval, and machine learning. ASTNN [77] employs an AST-based neural network that represents source code with smaller statement trees derived from segmented ASTs, capturing both lexical and syntactical knowledge. Finally, Sheneamer et al. [78] employ multiple machine learning classifiers to detect code clones, using features extracted from ASTs and PDGs. DLC [76] uses a recursive neural network approach to detect code similarity, utilizing the Euclidean distance as a metric.

Figs 17 and 18 present a comprehensive comparison of recall and F1 scores for each semantic clone between the proposed technique and the other selected techniques. The results for

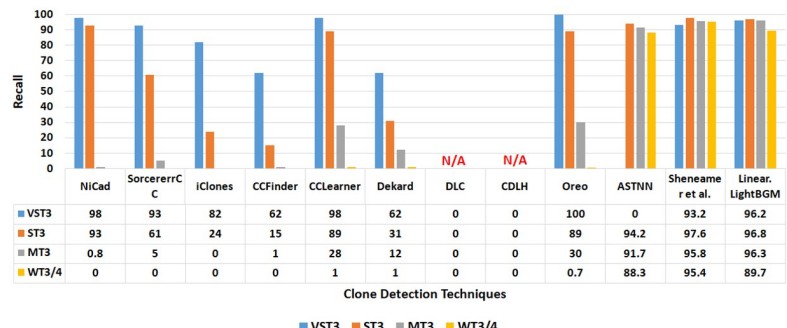

**Fig 17. Recall values when comparing the proposed method against selected methods with respected to different clone types.**

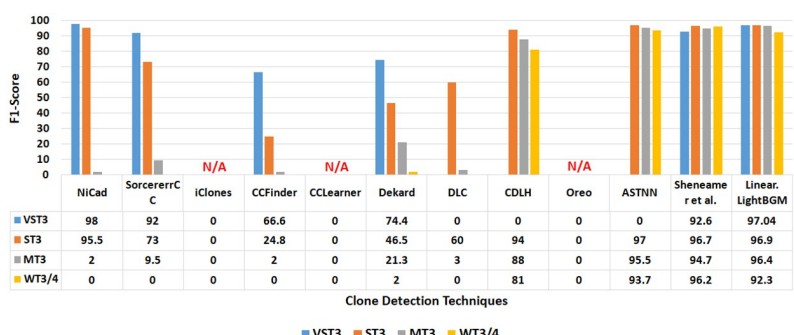

**Fig 18. F1-Score values when comparing the proposed method against selected methods with respected to different clone types.**

NiCad, SourcererCC, iClones, CCFinder, CCLearner, Dekard, Dlc, CDLH, Oreo, ASTNN, and Sheneamer et al. align with those reported in [7, 28, 31, 45, 75–78].

The results reveal low recall scores for NiCad, SorcerereCC, iClones, CCFinder, CCLearner, Dekard, and Oreo in identifying MT3 and WT3/4 clones, with scores of (0.8%,0%), (5%,0%), (0%,0%), (1%,0%), (28%,1%), (12%,1%), and (30%,0.7%) respectively. Additionally, NiCad, SourcererCC, CCFinder, Dekard, and DLC techniques exhibit low F1 scores for detecting MT-3 and WT3/4 clones, with scores of (2%,0%), (9.5%,0), (2%,0%), (21.3%,2%), and (3%,0%) respectively. Notably, F1 scores for CCLearner and Oreo are not reported and recall scores for DLC and CDLH are not reported. This indicates that these techniques may struggle to represent source code effectively, hindering their ability to capture comprehensive semantic features. Consequently, their performance (in terms of recall and F1 score) in detecting semantic clones, especially MT3 and WT3/4, is compromised.

In contrast, the proposed technique, employing the LightBGM classifier with a linear combination approach, outperforms NiCad, SourcererCC, CCFinder, CCLearner, iClones, Dekard, and Oreo in detecting the more complex clone types MT3 and WT3/4. It achieves higher recall and F1 scores, with values of (96.3%, 89.7%) and (96.4%, 92.3%) respectively. Furthermore, the proposed technique, utilizing the LightBGM classifier outperformed CDLH, ASTNN, and Sheneamer et al. [78] in detecting VST3, ST3, and MT3 clones in terms of F1-score.

The results shown in Figs 17 and 18 clearly demonstrate the effectiveness of the proposed code representation methodology in capturing comprehensive semantic features, thereby enhancing the identification of semantic clones.

## 6 Discussion

It is well-established that a higher level of abstraction in source code allows capturing a broader range of semantic code information [45]. This work is distinguished by its utilization of abstract features inherent in both high and low-level code representations [8, 41, 42]. Prior studies have concentrated on integrating these representations to identify Type-IV clones, emphasizing feature extraction from non-abstract representations. However, their approaches pose challenges due to the excessive details in non-abstract representations. What marks this study as distinctive is its utilization of the capabilities of static code analysis tools to convert programs into different abstract IRs, each offering distinct levels of abstraction and advantages for source code analysis, provides substantial advantages by eliminating extraneous elements while retaining the fundamental operations that convey the semantic meaning of the code fragment. These IRs diminish the effectiveness of obfuscation techniques, such as those that alter the syntactic structure while preserving the original code's semantics and functionality (e.g., metamorphism and polymorphism [26]).

This section summarizes the key findings and results obtained from the experiments and explores the following perspectives:

- **How well does the proposed approach perform in detecting different types of code clones?** Based on the results obtained from the experiments, it can be concluded that the proposed method shows promising performance in detecting code clones (syntactic and semantic). Ensemble classifiers, such as Random Forest, Rotation Forest, LightGBM, and Xgboost, achieved consistently high accuracy rates across different combinations, ranging from 95.65% to 96.75%, more details in Fig 10. Furthermore, the FeedForward Neural Network (FFNN) also demonstrated competitive performance, indicating its effectiveness in handling complex data relationships. These findings suggest that the proposed method, along with the selected classifiers, has the potential to effectively identify code clones in software.

- **What is the impact of integrating different features on the effectiveness of the proposed method in detecting code clones?** From the conducted experiments, it is evident that integrating different types of features, namely AST, BAF, and Jimple PDG, yields better results in detecting clones. Figs 11–13 illustrate the outcomes obtained by the highest-performing classifiers in these experiments, reaffirming the effectiveness of feature integration. Across various dataset sizes and combination techniques (linear, multiplicative, and distance), the classifiers consistently performed well. Particularly noteworthy is the observation that the linear combination method consistently outperformed the distance and multiplicative methods across all dataset sizes, with average performance disparities ranging from 3.6% to 7.4%. These findings emphasize the significance of feature integration in enhancing clone detection accuracy.

- **How does changing the number of features affect the effectiveness of the proposed approach in detecting code clones?** This study extends the previous research point by confirming the effectiveness of the proposed method in identifying code clones using varying numbers of features for each type. The results presented in Figs 14–16, obtained from top-performing classifiers, including Rotation Forest, Random Forest, LightGBM, and XgBoost, clearly show an upward trend in detection performance as the number of features increases.

This trend underscores the importance of the proposed features in achieving superior detection results, indicating that the observed improvements are not merely coincidental but rather driven by the inclusion of these features.

Based on the findings of this experiment, there is an opportunity to advance this research by investigating features of all types that contain more comprehensive semantic information and have made significant contributions to the semantic representation of code fragments. The techniques proposed by Heba and El-Hafeez [34, 35], which introduce novel techniques for selecting the most effective features, can be applied to identify the most important features that semantically encapsulate source code and reduce the overall number of features used.

- **How effective is the proposed method in detecting semantic clones?** The analysis of the effectiveness of the proposed code representation in detecting semantic clones, particularly those classified as VST3, ST3, MT3, and WT3/4, involved building five clone detection models. Results from Table 6 indicate that models utilizing Random Forest, Rotation Forest, and XGBoost classifiers show promising performance in identifying semantic clones. However, the model constructed using the LightGBM classifier outperforms others across all metrics, including F1-score, precision, and recall, for detecting semantic clones (VST3 F1-score: 97.04%, ST3 F1-score: 96.9%, MT3 F1-score: 96.4%, and WT3/4 F1-score: 92.3%). Overall, the employment of the proposed code representation yields commendable results in identifying semantic clones. This success can be primarily attributed to the effectiveness of the proposed code representation, which integrates source code features (AST) with features derived from abstract compiled IRs. This integration enables the capture of more comprehensive semantic features, thereby enhancing the efficiency of classifiers in identifying a broader range of Type-III and Type-IV clones.

- **How does the performance of the proposed technique in semantic clone detection compare to state-of-the-art approaches?** The analysis conducted in Section 5.2.5 demonstrates that the proposed approach for detecting syntactic and semantic clones outperforms several state-of-the-art clone detection techniques. Utilizing the LightBGM classifier with a linear combination approach, the proposed technique outperforms NiCad, SourcererCC, iClones, CCFinder, CCLearner, Dekard, DLC, CDLH, and Oreo in detecting complex clone types MT-3 and WT3/4, achieving higher recall and F1 scores. Specifically, it achieves recall and F1 scores of 96.3% and 96.4%, respectively, for MT3 clones, and 89.7% and 92.3%, respectively, for WT3/4 clones. Overall, the results clearly demonstrate the effectiveness of the proposed code representation in capturing more comprehensive semantics features, thereby improving the identification of semantic clones.

## 7 Threats to validity and limitations

There are four primary threats to the validity of the proposed approach.

Firstly, the effectiveness of the proposed approach is only demonstrated using the BigCloneBench dataset. However, it's important to note that BigCloneBench contains code fragments extracted from real-world Java repositories sourced from the IJaDataset 2.0. Moreover, BigCloneBench is widely recognized as a benchmark for evaluating code clone detection methods, thereby addressing this potential limitation.

Secondly, by focusing exclusively on method-level clones, there is a risk of overlooking overlapping clones or clones within Java classes. Nevertheless, it's worth noting that the majority of clones in Java code fragments occur at the method level [78].

The third threat arises from relying on previously published results from various baseline studies. This reliance was necessary due to unclear experiment settings in some baselines, like DLC and CDLH. However, depending directly on reported data from existing papers makes it difficult for us to conduct a more detailed qualitative comparison, including counting and analyzing each clone fragment between the proposed technique and others.

Finally, this study exclusively focuses on the Java programming language, chosen for its enduring prominence in the software development domain. Java language consistently ranks among the top five programming languages in the TIOBE index [79], indicating its continued relevance and widespread use. Moreover, Java's popularity has experienced a significant boost due to its critical role in Android application development.

## 8 Conclusion

This work presents a novel technique for detecting syntactic and semantic clones in Java source code, leveraging a unique code representation that integrates high-level source code features with low-level abstract compiled code features. High-level features are extracted from the AST of the source code, while the low-level features are derived from IRs—Baf, Jimple, and Jimple block PDG- generated by the Soot framework. Fifteen classifiers were trained and evaluated using this proposed code representation. Extensive experiments were conducted on the comprehensive real-world dataset, BigCloneBench, demonstrating the effectiveness of the approach in identifying semantic clones. The ensemble classifiers, including Random Forest, Rotation Forest, LightGBM, and XGBoost, achieved high accuracy rates across different classifiers, ranging from 95.65% to 96.75%. Moreover, the LightGBM classifier trained on the proposed technique, outperformed other baseline techniques, achieving higher recall and F1 scores of 96.3% and 96.4%, respectively, for MT3 clones, and 89.7% and 92.3%, respectively, for WT3/4 clones.

In future research, we plan to explore different deep learning architectures, including Recurrent Neural Networks (RNNs), Graph Convolutional Neural Networks (GCNNs), and Convolutional Neural Networks (CNNs), for semantic clone detection using this proposed code representation. Additionally, we will be considering extending this technique to detect semantic clones in programming languages beyond Java, with an assessment of its applicability across diverse codebases. Furthermore, we aim to investigate the effectiveness of the technique in detecting clones across various programming languages to expand its scope and impact on code clone detection. Moreover, scalability and other perspectives, such as performance optimization, were not investigated in this study. However, these areas offer promising opportunities for further investigation.

## Supporting information

**S1 File. List of features used in this study.**
(PDF)

**S2 File. Descriptive analysis of all clones types in BigCloneBench.**
(PDF)

**S3 File. Supplementary experiment results.**
(PDF)

**S4 File. Constructed datasets.**
(RAR)

## Acknowledgments

The authors express their gratitude for the support of University of Peshawar in proposing this work.

## Author Contributions

**Conceptualization:** Fahmi H. Quradaa, Sara Shahzad.

**Methodology:** Fahmi H. Quradaa, Sara Shahzad, Rashad Saeed.

**Project administration:** Fahmi H. Quradaa, Sara Shahzad.

**Software:** Rashad Saeed.

**Supervision:** Fahmi H. Quradaa, Sara Shahzad.

**Visualization:** Mubarak M. Sufyan.

**Writing – original draft:** Fahmi H. Quradaa.

**Writing – review & editing:** Rashad Saeed, Mubarak M. Sufyan.

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
