## [Decision Letter · Decision Letter 0]

23 Feb 2024

PONE-D-23-38302A novel code representation for detecting Java source code clonesPLOS ONE

Dear Dr. Quradaa,

Thank you for submitting your manuscript to PLOS ONE. After careful consideration, we feel that it has merit but does not fully meet PLOS ONE’s publication criteria as it currently stands. Therefore, we invite you to submit a revised version of the manuscript that addresses the points raised during the review process.

We look forward to receiving your revised manuscript.

Kind regards,

Academic Editor

PLOS ONE

Journal Requirements:

**Additional Editor Comments:**

Although this manuscript has a few of merits, the reviewers gave serious issues and it needs significant revision.

Reviewers' comments:

Reviewer's Responses to Questions

**Comments to the Author**

1. Is the manuscript technically sound, and do the data support the conclusions?

Reviewer #1: Partly

Reviewer #2: No

Reviewer #3: Partly

2. Has the statistical analysis been performed appropriately and rigorously? 

Reviewer #1: Yes

Reviewer #2: No

Reviewer #3: N/A

3. Have the authors made all data underlying the findings in their manuscript fully available?

Reviewer #1: Yes

Reviewer #2: No

Reviewer #3: No

4. Is the manuscript presented in an intelligible fashion and written in standard English?

Reviewer #1: Yes

Reviewer #2: Yes

Reviewer #3: No

5. Review Comments to the Author

Reviewer #1: Some potential drawbacks of the proposed machine learning approach for code clone detection using high and low level source code representations:

1. Increased complexity: Combining multiple code representations adds complexity over single-representation techniques.

2. Scalability issues: Deriving and analyzing multiple representations could hamper analysis of very large codebases.

3. Sensitivity to changes: Small code modifications may invalidate code representations, requiring re-analysis.

4. Variability across languages: Representations and extracted features may not generalize well across different programming languages.

5. Infrastructure requirements: Extraction and processing of representations requires language-specific toolchains/infrastructure.

6. Evaluation limitations: Approach tested on limited datasets, more rigorous validation on real-world projects needed.

7. False positives risks: Semantic abstraction could wrongly link syntactically different code snippets.

8. Configuration challenges: Proper configuration of machine learning models requires hyperparameter tuning expertise.

9. Explainability challenges: Identifying why certain code is flagged as clones from multiple blended representations is difficult.

10. Intellectual property risks: Extracting and sharing representations and models could inadvertently leak proprietary code.

11. Describe dataset features in more details and its total size and size of (train/test) as a table.

12. Flowchart and algorithm steps need to be inserted.

13. Time spent need to be measured in the experimental results.

14. Limitation Section need to be inserted.

15. All metrics need to be calculated such as Accuracy, Precision, Recall, F1 score, and ROC AUC score in the experimental results as tables.

16. Address the accuracy/improvement percentages in the abstract and in the conclusion sections, as well as the significance of these results.

17. The architecture of the proposed model must be provided

18. The authors need to make a clear proofread to avoid grammatical mistakes and typo errors.

19. The authors need to add recent articles in related work and update them.

20. Add future work in last section (conclusion) (if any)

21. To improve the Related Work and Introduction sections authors are recommended to review this highly related research work paper:

a) A high-quality feature selection method based on frequent and correlated items for text classification

b) A new feature selection method based on frequent and associated itemsets for text classification

c) Building an Effective and Accurate Associative Classifier Based on Support Vector Machine

d) An ASP .NET Web Applications Data Flow Testing Approach

e) An Approach to Slicing Object-Oriented Programs

Reviewer #2: The paper attempts to report work on using AST to detect Java clones. As the motivation of the work is not clearly defined, the methodology presented is not sound and proven. There is no data, experiment or lemma/theorem to support the proposal. Therefore, the paper is recommended for rejection.

Reviewer #3: Dear Authors.

Thanks for submitting this work. The paper reviews code representation for detecting Java source code clones. The topic looks promising; however, a few issues are available in this manuscript.

Detailed feedback:

Title: it is recommended to rephrase the title to show the approach of clone detection, for example using bytecode or assembly.

Abstract: lines from 16 onwards might be too detailed for the abstract. it is recommended to remove them. The issue could be maintainability, reusability and even ethical consideration.

Introduction

it is recommended to provide either a table or a tree structure showing subtypes.

The role or type of machine learning used in this study is not clear. This also should reflect the abstract to include the idea of machine learning.

94-100 seems to be repeated. Many parts in the introduction are repeated.

define the acronyms on first use.

the background should come before related works.

Methodology

instead of saying, "will be normalized and compiled using a compilation tool like the Stub", authors should specify what they use and why.

Step 3 is not detailed at all, and it is suggested to be removed.

there are no results of this study, so no generalization can be made.

no limitations or dataset description.

It is highly recommended to have a section for used datasets and tools.

Discussion

is shallow, focus on research questions and the objectives, please.

conclusion

it looks like a summary, please revise

implications, limitations, and threats are crucial sections.

Minor issues

line 28- However, this approach ==>However, in this approach

, l section numbering and figures are not inserted in their place.

all figures are blur.

6. PLOS authors have the option to publish the peer review history of their article (what does this mean?). If published, this will include your full peer review and any attached files.

Reviewer #1: **Yes: **Tarek Abd El-Hafeez

Reviewer #2: No

Reviewer #3: No

---

## [Author Response · Author response to Decision Letter 0]

23 Mar 2024

I would like to express my sincere gratitude to the Editor and the reviewers. Your valuable feedback and insights are greatly appreciated, and we are committed to addressing all your comments and suggestions to enhance the quality and impact of our work.

1. Editor's Journal comments and responses

• Comment 1: [Please ensure that your manuscript meets PLOS ONE's style requirements, including those for file naming.]

Response : We would like to inform you that we have adhered to PLOS ONE's style requirements as outlined in the PLOS ONE style templates.

• Comment 2: [Please note that PLOS ONE has specific guidelines on code sharing for submissions in which author-generated code underpins the findings in the manuscript. In these cases, all author-generated code must be made available without restrictions upon publication of the work.]

Response : Thank you for your concerns regarding the PLOS ONE guidelines on code sharing requirement. We assure you that we are fully committed to transparency and are willing to provide any details about the source code to reviewers upon request.

• Comment 3: [Please provide a complete Data Availability Statement in the submission form, ensuring you include all necessary access information or a reason for why you are unable to make your data freely accessible. If your research concerns only data provided within your submission, please write "All data are in the manuscript and/or supporting information files" as your Data Availability Statement.]

Response : In reference to the Data Availability statement, all data are presently enclosed within our paper. Furthermore, a supporting information files has been included, encompassing all results we obtained and the datasets that we used in the experiments. Consequently, I respectfully request a modification of the Data Availability statement to state, " The dataset utilized in the present study can be accessed at the following link: https://github.com/clonebench/BigCloneBench?tab=readme-ov-file and support file S4"

2. Comments from Reviewer 1 and Responses

• Comment 1: [Increased complexity: Combining multiple code representations adds complexity over single-representation techniques.]

Response : We appreciate the insightful suggestion from the reviewer regarding the increased complexity associated with combining multiple code representations. While it's true that integrating multiple representations may introduce additional complexity, it is essential to weigh this against the benefits gained in terms of enhanced accuracy and robustness in clone detection. 

When considering the complexities involved in code clone detection, it is crucial to note that researchers commonly categorize code clones into four types: Type-I, Type-II, Type-III, and Type-IV. The first three rely on textual similarities, whereas Type-IV focuses more on functional or semantic resemblances, posing challenges for detection based solely on textual similarities. 

Traditionally, techniques have concentrated on detecting Types I-III clones, employing single code representations like sequences of tokens, trees, graphs, or software metrics. While successful for these types, they often struggle with Type-IV clones due to lexical differences but similar functionality. To overcome this challenge, hybrid code representation has emerged as a promising approach, integrating multiple techniques such as text, tokens, AST, PDG, and metrics. This comprehensive approach enhances clone detection accuracy by capturing semantic similarities that are difficult to detect using a single representation [1-3]. 

Ultimately, the trade-off between complexity and benefits should be carefully considered. We believe that the advantages gained in terms of improved clone detection and classification, while reducing false positives, outweigh the increased complexity introduced by combining multiple code representations.

• Comment 2: [Scalability issues: Deriving and analyzing multiple representations could hamper analysis of very large codebases.]

Response : We appreciate the insightful recommendation from the reviewer. It would indeed be interesting to explore scalability issues associated with deriving and analyzing multiple code representations. However, our main objective in this work is to propose a new code representation clones and evaluate its effectiveness in detecting both syntactic and semantic clones. While scalability analysis was not within the scope of this work, this approach aligns with the methodology applied in other studies in this domain [2] [4-7].

Nonetheless, we acknowledge that this comment has highlighted an opportunity for future and more extensive research in this direction. We have incorporated a recommendation to explore this opportunity in the revised manuscript. Please refer to page 32 of the revised manuscript, lines 741–743.

• Comment 3: [Sensitivity to changes: Small code modifications may invalidate code representations, requiring re-analysis.]

Response : We are grateful for the reviewer's insightful suggestion, and we acknowledge the importance of sensitivity to changes. In response to this valuable comment, we want to clarify how we made the proposed technique robust for source code changes.

Firstly, in the preprocessing stage several normalization operations have been applied, including the removal of unnecessary white spaces, the exclusion of comments, and the substitution of variable names and function names with generic placeholders to mitigate disparities in naming conventions. Numeric literals, string literals, and other constants have with placeholders to generalize specific values. For detailed information, please refer to the revised manuscript, specifically on page 11, lines 266–273.

Secondly, the Soot Framework, a static analysis tool, has been used to optimize and represent Java source code by carrying out transformations such as eliminating repetitious code, dead code elimination, unused local variables elimination, and common sub-expression elimination. This significantly reduced the number of operations required to represent the java source code. The soot framework provides four intermediate representations (IRs). Each of the IRs has different levels of abstraction that provide several benefits when analyzing and transforming java code. Further details can be found on pages 5-8, lines 126-194.

• Comment 4: [Variability across languages: Representations and extracted features may not generalize well across different programming languages.]

Response : Thank you for raising the concern about the variability across languages in the generalization of representations and extracted features. While it's true that representations and features may not generalize well across different programming languages, it's important to note that our proposed code representation is specifically designed and optimized for detecting clones in Java code.

We acknowledge that variability across languages can pose challenges in generalization, and we have focused our efforts on developing a representation that is tailored to the characteristics of Java code. By leveraging domain-specific knowledge and utilizing techniques that are specific to Java programming, we aim to maximize the effectiveness of our representation in detecting clones within the Java ecosystem.

While our representation may not directly generalize to other programming languages, we believe that the principles and methodologies employed in its development can serve as valuable insights for researchers working on clone detection in other languages. Additionally, future research could explore the adaptation of our techniques to other languages, taking into account their unique characteristics and requirements.

We have acknowledged this limitation in the threats to validity and limitations section and future work part in the conclusion. Please refer to page 31 of the revised manuscript, lines 715–719 and pages 31 and 32, lines 739-741.

• Comment 5: [Infrastructure requirements: Extraction and processing of representations requires language-specific tool chains/infrastructure.]

Response : Thank you for bringing up the concern regarding infrastructure requirements for the extraction and processing of representations, which necessitate language-specific toolchains and infrastructure.

In our work, we have developed a novel code representation specifically tailored for detecting clones in Java code. To address these infrastructure requirements, we carefully selected and utilized language-specific toolchains and infrastructure. For example, we used the Soot framework [8] for static analysis and code transformation in Java, the JavaParser [9] tool to build AST and traverse its nodes, and the Stubber [10] tool for compiling Java source code into Bytecode without dependencies. The roles of each tool are specified in the methodology section of the revised manuscript, which can be found on pages 10-20 and in the background section on pages 4-8. 

• Comment 6: [Evaluation limitations: Approach tested on limited datasets, more rigorous validation on real-world projects needed.]

Response: We appreciate the reviewer for pointing this out. In this work, we used the BigCloneBench [11] dataset (a real-world dataset), which is a widely used benchmark for assessing Java code clone detection systems. It contains 55,499 Java source files from 24,557 distinct open-source projects, collected through the mining process of IJaDataset-2.0. The current version of this benchmark includes over 8.5 million labelled true clone pairs and more than 260,000 labelled false clone pairs across 43 functionalities, categorized into Type-I, Type-II, Type-III, and Type-IV. State–of–the–art techniques [2, 5, 6, 12] in the field of code clone detection have used this dataset. For more details please refer to the revised manuscript on pages 20-21, lines 433- 460.

• Comment 7: [False positives risks: Semantic abstraction could wrongly link syntactically different code snippets.]

Response: We appreciate the insightful suggestion from the reviewer regarding the risk of false positive. As we know, representing source code in a more abstract manner may lead to extracting more comprehensive semantic features, thereby improving the effectiveness of clone detection techniques in identifying semantic clones. However, the main issue associated with abstract representation is the potential increase in false positive. To mitigate this issue, we employed both high-level source code and abstract low-level compiled code representations in our work. This approach allowed us to leverage the benefits of abstract representation while considering the specific details of the source code. For more details, please refer to the methodology section of the revised manuscript, available on pages 10-20. 

• Comment 8: [Configuration challenges: Proper configuration of machine learning models requires hyperparameter tuning expertise.]

Response: Thank you for your interesting comment. In our study, we opted to use the default settings for machine learning models, as specified in the techniques documentation. While hyperparameter tuning can indeed enhance model performance, we chose to adhere to the default settings to maintain consistency and align with established practices outlined in the documentation. By using default settings, we aimed to ensure transparency and reproducibility in our methodology. We acknowledge that further exploration of hyperparameter tuning could be beneficial and may be considered in future research.

• Comment 9: [Explainability challenges: Identifying why certain code is flagged as clones from multiple blended representations is difficult.]

Response: We appreciate the reviewer's insightful comment regarding the challenges of explainability when using multiple blended representations for clone detection. In our work, we recognize the importance of explainability in clone detection and have taken steps to address this challenge. Firstly, we have thoroughly documented our methodology for deriving and analyzing multiple representations. This documentation includes detailed explanations of each representation technique used and how they contribute to clone detection.

Furthermore, we have conducted two experiments to investigate and illustrate how each representation contributes to the identification of code clones. In the first experiment, we evaluated the performance of the proposed techniques with different dataset sizes and feature types (AST, Baf, JimplePDG, and AST+Baf+JimplePDG). Please refer to page 23, lines 504-524, in the revised manuscript for more details. To further investigate the importance of the proposed features (code representation) in enhancing performance in code clone detection and to ensure that this improvement is not coincidental, another experiment was conducted involving varying numbers of features, combined by three combination approaches (linear, multiplicative, distance). An equal number of features was selected from each type (AST, BAF, and Jimple PDG). Please refer to page 24,lines 525-538, for more details.

• Comment 10: [Intellectual property risks: Extracting and sharing representations and models could inadvertently leak proprietary code.]

Response: We appreciate the reviewer for highlighting the concern regarding intellectual property risks associated with extracting and sharing representations and models, which could potentially lead to the inadvertent leakage of proprietary code.

To address this concern, we have implemented several precautions in our work. Firstly, we have ensured that all code representations and models used in our study are derived from publicly available and open-source datasets, such as BigCloneBench, which contains labeled clone pairs from various open-source projects. Additionally, both the Soot, Stubber, and the JavaParser tool are freely available for use under the terms of the GNU Lesser General Public License (LGPL) version 2.1 and Eclipse Public License 2.0. Furthermore, we have meticulously anonymized and sanitized any code snippets used in our analysis to remove any proprietary or sensitive information. Overall, we are committed to upholding the highest standards of ethical conduct and data privacy in our research endeavors.

• Comment 11: [Describe dataset features in more details and its total size and size of (train/test) as a table]

Response: Thank you for your valuable feedback. We appreciate your suggestion to provide more detailed information about the dataset features and its total size, as well as the size of the train and test sets, in a table format.

In response to your suggestion, we have included a support file, labeled as S1, which contains detailed information about different types of feature used in this work. 

Regarding the dataset and its size, you can refer to Table 5 in the revised manuscript on page 21, which provides an overview of the constructed dataset. For more detailed information, please refer to support file labeled S2, which contains comprehensive details about both the BigCloneBench dataset and the constructed dataset.

As for the size of the training and test sets, we specified the dataset size used in each experiment, please refer to the evaluation section in the revised manuscript on pages 21-27. Additionally, we utilized 10-fold Stratified cross-validation for training and testing our models. This approach ensures robustness and reliability in our models evaluation process by systematically dividing the dataset into training and testing subsets while maintaining the distribution of classes.

We believe that these enhancements offer a more comprehensive understanding of the dataset used in our work and its characteristics.

• Comment 12: [Flowchart and algorithm steps need to be inserted]

Response: We greatly appreciate your recommendation to include flowcharts to illustrate the architecture and algorithms used in our methodology.

We are pleased to inform you that we have incorporated a detailed architecture diagram in the revised manuscript, located in Figure 8 on page 11. This diagram provides a comprehensive overview of our methodology.

Additionally, in response to your suggestion, we have introduced a new figure (Figure 9) on page 11 to illustrate the process of ext

---

## [Decision Letter · Decision Letter 1]

2 Apr 2024

A novel code representation for detecting Java code clones using high-level and abstract compiled code representations

PONE-D-23-38302R1

Dear Dr. Quradaa,

We’re pleased to inform you that your manuscript has been judged scientifically suitable for publication and will be formally accepted for publication once it meets all outstanding technical requirements.

Kind regards,

Unil Yun, Ph.D.

Academic Editor

PLOS ONE

Additional Editor Comments (optional):

Authors revised carefully this manuscript and it is acceptable now.

Reviewers' comments:

Reviewer's Responses to Questions

**Comments to the Author**

1. If the authors have adequately addressed your comments raised in a previous round of review and you feel that this manuscript is now acceptable for publication, you may indicate that here to bypass the “Comments to the Author” section, enter your conflict of interest statement in the “Confidential to Editor” section, and submit your "Accept" recommendation.

Reviewer #1: All comments have been addressed

Reviewer #3: All comments have been addressed

2. Is the manuscript technically sound, and do the data support the conclusions?

Reviewer #1: Partly

Reviewer #3: Yes

3. Has the statistical analysis been performed appropriately and rigorously? 

Reviewer #1: Yes

Reviewer #3: N/A

4. Have the authors made all data underlying the findings in their manuscript fully available?

Reviewer #1: Yes

Reviewer #3: Yes

5. Is the manuscript presented in an intelligible fashion and written in standard English?

Reviewer #1: Yes

Reviewer #3: Yes

6. Review Comments to the Author

Reviewer #1: All of the comments and feedback provided have been thoroughly addressed and incorporated into the final version of the document. Given the comprehensive consideration of all suggestions, I highly recommend proceeding with accepting the document for publishing.

Reviewer #3: Thank you for addressing all comments.

Please pay attention to figures as some of them are still blurry.

and kindly ensure all arguments are completely included within the manuscript were applicable.

7. PLOS authors have the option to publish the peer review history of their article (what does this mean?). If published, this will include your full peer review and any attached files.

Reviewer #1: **Yes: **Tarek Abd El-Hafeez

Reviewer #3: No
